# Onset of Action of Bupivacaine Administered via Dural Puncture Epidural versus Spinal, Lumbosacral, and Sacrococcygeal Epidural Injections in Dogs: Randomised Clinical Trial

**DOI:** 10.3390/ani11102996

**Published:** 2021-10-19

**Authors:** Fernando Martinez-Taboada, Tsim Christopher Sun, Jose Ignacio Redondo

**Affiliations:** 1Sydney School of Veterinary Sciences, University of Sydney, Sydney, NSW 2050, Australia; christopher.sun@sydney.edu.au; 2Department of Animal Medicine and Surgery, Faculty of Veterinary Medicine, University of Murcia, 30100 Murcia, Spain; 3Departamento de Medicina y Cirugía Animal, Facultad de Veterinaria, Universidad CEU Cardenal Herrera, 46009 Valencia, Spain; nacho@uchceu.es

**Keywords:** epidural anaesthesia, spinal anaesthesia, dural puncture epidural, onset, dog, bupivacaine

## Abstract

**Simple Summary:**

Epidural anaesthesia is commonly performed to provide long-lasting local anaesthesia in animals. Spinal anaesthesia has become popular in human anaesthesia due to a faster onset, but shorter duration, than epidural anaesthesia. Recently, dural puncture epidural anaesthesia (DPE) has been proposed as a compromise between spinal and epidural anaesthesia, providing longer duration than spinal anaesthesia with similar onset of action. This study aimed to compare the technical aspects of these techniques (onset of action, number of attempts, and time to perform them). The results showed no difference between the techniques in number of attempts required, while the epidural injections tended to be shorter than spinal and dural puncture epidural injections. The onset of action (measured as disappearance of the patellar reflex) was longer in the sacrococcygeal epidural injection than any other techniques (including lumbosacral epidural injections).

**Abstract:**

The study aimed to compare bupivacaine onset time when administered via epidural anaesthesia injecting both at the lumbosacral and sacrococcygeal spaces, spinal anaesthesia, and DPE in clinical dogs. A total of 41 dogs requiring neuraxial anaesthesia as part of their anaesthetic protocol were recruited. They were randomly allocated to receive an epidural injection in the sacrococcygeal space aided by the nerve stimulator (SCO), an epidural injection in the lumbosacral (LS), a subarachnoid injection (SPI), or a DPE. The onset of anaesthesia was assessed every 30 s after the injection by testing the presence of patellar ligament reflex. The number of attempts and time to perform the technique were also recorded. Data were analysed using a one-way ANOVA for trimmed means with post hoc Lincoln test and a Kaplan–Meier curve. The significance level was set at *p* < 0.05, and the results are presented in absolute values and median (range). There was no difference in the number of attempts required to complete the techniques between groups (*p* = 0.97). Epidural injections (LS and SCO) tended to be shorter than SPI and DPE techniques, but there was no statistically significant difference (*p* = 0.071). The time to the disappearance of patellar ligament reflex (Westphal’s sign) in the SCO group was longer than in any other group. In conclusion, all techniques provided a rapid block of the patellar reflex. The SCO technique was the slowest in onset, while the other groups (SPI, DPE, and LS) were faster and almost indistinguishable.

## 1. Introduction

Epidural anaesthesia is still one of the most used regional anaesthetic techniques in small animals. Its versatility allows the desensitisation of the tail, perineal area, pelvic limbs, abdomen, or thorax only, by the adjustment of the drugs injected, their doses, and site of injection [1]. In dogs, epidural anaesthesia has been proven to provide a reliable and superior pain relief compared to parenteral opioids in stifle surgeries [2] and to block the neuroendocrine stress response induced by the surgical insult [3]. The most common site for epidural injections in small animals has traditionally been the lumbosacral space [1,4]. In recent years, the confirmation that the dural sac in cats extends caudal to the lumbosacral and that it is readily damaged during the epidural injection [5], as well as the suspicion that this may also occur in dogs [6,7], have prompted researchers to propose the sacrococcygeal space as an alternative injection site [8,9,10,11,12].

Spinal anaesthesia is extensively used in humans, because it has certain advantages over epidural anaesthesia. From a technical point of view, spinal anaesthesia provides an objective end-point (presence of cerebrospinal fluid in the hub of the needle), but no method of identification of the epidural space can guarantee the location of the needle tip in all patients and conditions [13]. Other advantages are a faster onset of action, a more controllable and predictable spread of anaesthesia, and a more profound degree of surgical anaesthesia [13]. Contrarily, the disadvantages of spinal anaesthesia are a shorter duration of action, greater haemodynamic impact, and a greater risk of post dural puncture headache (PDPH) than epidural injection [13]. The veterinary literature regarding spinal anaesthesia is very limited and only covers some of these aspects of the technique. Sarotti et al. demonstrated that spinal anaesthesia provided a better intraoperative analgesia and faster recovery of the motor function than epidural anaesthesia, with a considerably lower procedural failure rate [14]. Unfortunately, there are no reports about other aspects such as the onset of action or the likelihood of PDPH.

Dural puncture epidural (DPE) refers to the puncture of the dura mater with a spinal needle, without spinal administration of any drug, prior to the epidural injection of drugs. This technique has been increasing in popularity in human anaesthesia for the last two decades as a modification of the combined spinal–epidural anaesthesia. DPE maintains some of the advantage of spinal and epidural anaesthesia such as the rapid onset of action, long duration and profound level of analgesia minimising unwanted effects, especially haemodynamic instability and PDPH [15,16].

The aim of this study was to compare DPE, epidural anaesthesia (both at lumbosacral and sacrococcygeal intervertebral spaces), and spinal anaesthesia in clinical dogs with a focus on the technical aspects of these techniques (success rate, time to perform the procedure, number of attempts, and onset of action). The hypothesis was that epidural anaesthesia would be the fastest technique requiring fewer attempts, but with the longest onset of action.

## 2. Materials and Methods

The study was approved by the Animal Ethics Committee of the University of Sydney (2018/1427), and informed owner consent was obtained before inclusion in this research. The forty-one dogs recruited for this study presented at the University Veterinary Teaching Hospital of the University of Sydney for surgical procedures involving the pelvic limbs requiring neuraxial anaesthesia with bupivacaine as part of the individualised balanced anaesthetic protocol. All the animals were considered healthy or with a mild systemic disease (American Society of Anaesthesiologist Physical Status (ASA-PS) Classification I or II) based on the evaluation of the clinical history, a thorough physical examination, and basic haematology and biochemistry. The exclusion criteria were animals with an ASA-PS classification above II, unpalpable anatomical landmarks, spinal anatomical abnormalities, cutaneous infections in the area of interest, and those with no detectable patellar reflex prior to the neuraxial technique.

The dogs’ anaesthetic protocol was individualised by the attending anaesthetist based on the temperament of the animal, the ASA-PS, the surgical procedure, and personal preferences. All animals were sedated intramuscularly with a combination of methadone (0.2–0.3 mg/kg; Methadone 10 mg/mL injection Ilium, Troy Laboratories Pty Ltd., Glendenning, NSW, Australia), and one or more of the following sedatives: medetomidine (5–10 µg/kg; Medetomidine injection Ilium, Troy Laboratories Pty Ltd., Glendenning, NSW, Australia), dexmedetomidine (3–5 µg/kg; Dexdomitor, Jurox Pty Ltd., Rutherford, NSW, Australia), acepromazine (0.02–0.03 mg/kg; A.C.P. 2, Ceva Animal Health Pty Ltd., Glenorie, NSW, Australia), and ketamine (0.5–1 mg/kg; Ketamine 100 injection Ilium, Troy Laboratories Pty Ltd.). After 15–30 min, induction of anaesthesia was performed intravenously with either propofol (Propofol Lipuro, B. Braun Mesungen AG, Mesungen, Germany), alfaxalone (Alfaxan Multidose, Jurox Pty Ltd., Rutherford, NSW, Australia), or the combination of one of these drugs with midazolam (Hypnovel 5 mg/mL, Roche Products Pty Ltd., Sydney, NSW, Australia), ketamine (Ketamine 100 injection Ilium, Troy Laboratories Pty Ltd.), or lidocaine (Lignocaine injection 2%, Pfizer Perth Pty Ltd., Bentley, WA, Australia). When orotracheal intubation was completed, anaesthesia was maintained in all cases with isoflurane (VCA ISO Inhalation Anaesthetic, Veterinary Companies of Australia Pty Ltd., Sydney, NSW, Australia) vaporised in oxygen via a circle breathing system. Depth of anaesthesia and physiological parameters (electrocardiography, pulse oximetry, end-tidal carbon dioxide, oscillometric arterial blood pressure, and oesophageal temperature) were assessed continuously and recorded every five minutes (Anitek C50-V, Anitek Pty Ltd., Sydney, NSW, Australia). Hartmann’s solution (Baxter Viaflex Compound Sodium Lactate, Baxter Healthcare Pty Ltd., Sydney, NSW, Australia) was started at a rate of 5 mL/kg/h and adjusted accordingly by the attending anaesthetist.

After 10 min of stabilisation, the dogs were randomised into four study groups using an electronically generated list (https://studyrandomizer.com (accessed on 11 August 2020); Phase Locked Software, Wageningen, The Netherlands). Based on the study group, the animals were placed in sternal or lateral recumbency with the pelvic limbs pulled forward and the attending anaesthetist confirmed the presence of patellar reflex or knee-jerk using a standard Taylor’s reflex hammer on both pelvic limbs. Then, one of the following techniques was performed.

### 2.1. Sacrococcygeal Epidural Anaesthesia (SCO)

With the animal in sternal recumbency, the hair over the area of interest was clipped and the skin was surgically prepared. The sacrococcygeal intervertebral space was identified immediately caudal to the last spinous process of the median sacral crest, and it was confirmed by the dorsoventral flexion of the tail base. A 22-gauge insulated needle (Stimuplex needle 0.7 × 50 mm, B. Braun, Melsungen AG, Germany) connected with the negative lead to a nerve stimulator (EZstim II, Life-Tech Inc., Houston, TX, USA) and the positive connected to the left foot of the animal was used. The stimulator was set to deliver a constant current of 0.7 mA at a frequency of 2 Hz and with a pulse width of 0.1 ms. The step-by-step technique that was followed to identify the sacrococcygeal epidural space using the nerve stimulator has been previously published elsewhere [11]. Immediately after needle position, presence of blood was ruled out by gently aspirating with a test syringe. The dose of bupivacaine (Bupivacaine injection BP 0.5%, Pfizer Perth Pty Ltd., Bentley, WA, Australia) was calculated as 0.05 mL/cm occipital–coccygeal length and mixed with buprenorphine (4 µg/kg; Temgesic injection, Indivior Pty Ltd., Ryde, NSW, Australia) prior to the epidural injection over 30 s [11].

### 2.2. Spinal Anaesthesia (SPI)

With the animal in left lateral recumbency, the hair over the lumbar vertebrae L4 to L7 was clipped and the skin surgically prepared. A 22-gauge Quincke spinal needle (63 to 75 mm length depending of the size of the dog) (BD Spinal needle, Spinal needle Quincke type point, Becton Dickinson SA, Madrid, Spain) was inserted in a parasagittal approach between the lumbar vertebrae L5 and L6 until cerebrospinal fluid was obtained, as previously described by Sarotti et al. [14]. The dose of bupivacaine was calculated as 0.21 mg/kg body weight + 0.035 mg/cm occipital coccygeal length [14], and then mixed with buprenorphine (1 µg/kg) before being injected in the subarachnoid space over 30 s.

### 2.3. Lumbosacral Epidural Anaesthesia (LS)

With the animal in sternal recumbency, and after clipping the hair and surgically cleaned the area of interest, a 22 gauge Quincke spinal needle (63 to 75 mm length) was introduced in the lumbosacral intervertebral space. The epidural space was identified using the hanging drop technique and/or the lack of resistance to the injection prior to the final injection of the bupivacaine and buprenorphine combination. The drug doses of bupivacaine and buprenorphine were calculated as in the SCO group and administered over 30 s, and the local technique followed the steps previously reported in the literature [17]. The epidural injection was delivered once the presence of blood or cerebrospinal fluid was ruled out via a gentle aspiration with a syringe.

### 2.4. Dural Puncture Epidural Anaesthesia (DPE)

With the animal initially in lateral recumbency, the skin over the area between the L5 vertebra and the sacrococcygeal area was clipped and sterilely prepared. A spinal puncture was performed in the L5-L6 intervertebral space in the same way as in the SPI group, but no drug was administered after obtaining several drops of cerebrospinal fluid through the Quincke spinal needle. After the dural puncture was completed and the needle fully removed from the animal, the dog was rolled into sternal recumbency, and the epidural injection was performed in the sacrococcygeal intervertebral space following the same protocol and dosing as in the SCO group.

In all dogs, and immediately after completing the bupivacaine and buprenorphine injections, the animals were positioned in left lateral recumbency (if they were not already positioned in that recumbency), and they were covered with a blanket for warmth and blinding purposes. The attending anaesthetist, unaware of the study group, assessed the presence of the patellar reflex on the right pelvic limb once every 30 s from the completion of the neuraxial injection until the abolition of this reflex until patellar reflex suppression (or Westphal’s sign) was recorded as onset of action of bupivacaine. In the case of obtaining a weak response to the patellar ligament hit, the response was considered positive, and the assessment was repeated 30 s later, until a negative response was obtained.

In all cases, the total number of attempts and the time to perform the technique were also recorded. The time taken to perform the technique was counted in seconds with a chronometer from the time the needle was rested on the animal’s skin, until the moment the anatomical target (either epidural or subarachnoid space) was reached. An attempt was considered to be any single forward movement of the needle, so two attempts were counted if the needle had to be withdrawn and redirected after hitting bone, for instance. In the DPE group, the chronometer was stopped when the subarachnoid space was pierced and restarted after the animal was repositioned and the needle was resting on the skin again to perform the epidural injection. Additionally, in this group, the number of attempts for each of the techniques performed were counted independently and only the highest number of attempts was used for statistical analysis. For example, in the case of requiring three attempts to perform the dural puncture and only one to complete the epidural injection, three attempts were used for statistical analysis.

After the period of data collection, the anaesthetic proceeded normally, being managed by the attending anaesthetist of the case, who had total freedom to adapt the anaesthetic protocol to the clinical circumstances of the case. At the end of surgery, when the animal was in the recovery room, the attending anaesthetist was then asked if, in their clinical opinion, the neuraxial technique had been successful (with the only possible answer being yes or no).

### 2.5. Data Analysis

A sample of at least 40 dogs (a minimum of 10 animals per group) was calculated as adequate for the detection of a one minute difference in the time to perform the technique with a level of significance of 5% and a power of 80%, using 2.56 min as the population variance (https://www.fisterra.com/mbe/investiga/9muestras/9muestras2.asp (accessed on 11 August 2020); Fisterra, Elsevier España S.L.U., Spain).

The data was analysed using an R 3.5.3 (R Core Team 2019, The R Foundation for Statistical Computing http://www.R-project.org (accessed on 25 July 2021), Austria). Sex was compared between groups using a Chi-Square test. Age, weight, BCS, OCL, total volume of local anaesthetic injected, the times to perform the technique and to Westphal’s sign, and the number of attempts were compared between groups using a one-way ANOVA for trimmed means using the function t1 way () of the package WRS2 [18]. The post hoc analysis was performed with the Lincoln test (function lincon ()), also in the WRS2 package. The surfit () from the survival package [19] was used to analyse the time to Westphal’s sign in all dogs in the four study groups without controlling the rest of the factors; then, the Kaplan–Meier ‘survival’ curves were plotted. In all the analyses, the level of significance was always set at *p* < 0.05. Data related to the anaesthetic drugs, doses, and animal signalment were analysed with descriptive statistics and presented as mean ± SD, median (range), and/or number of observations, as required.

## 3. Results

Eventually, a total of 41 dogs were included in the study, and no animals dropped out or were excluded. The dogs were scheduled for a variety of orthopaedic and soft tissue surgeries (Table 1), and animals’ signalments were similar between groups, apart from the SPI group having an overrepresentation of male dogs compared to the rest of the groups (Table 2).

Multiple drugs combinations were used for sedation and induction of anaesthesia (Table 3). All the injections were completed without any complication. The volume injected in the SPI group was considerably smaller than the volumes injected in the other study groups (Table 2).

There was no difference between study groups in the number of attempts used to complete the techniques, (*p* = 0.97) with the number of attempts being 2 (1–4), 2 (1–3), 2 (1–3) and 2 (1–3) for the DPE, LS, SCO, and SPI groups, respectively (Figure 1). The epidural injections (LS and SCO groups) tended to be consistently shorter (34 (10–120) and 46 (26–210) s, respectively) than the DPE and SPI techniques (110 (70–317) and 112 (16–447) s, respectively), but the differences were not statistically significant (DPE vs. LS: *p* = 0.21, CI95%: -36–258; DPE vs. SCO: *p* = 0.27, CI95%: 246–(-48); DPE vs. SPI: *p* = 0.56, CI95%: 203–(-114); LS vs. SCO: *p* = 0.56, CI95%: 48–(-72); LS vs. SPI: *p* = 0.29, CI95%: 39–(-172); SCO vs. SPI: *p* = 0.40, CI95%: 52–(-161)) (Figure 2).

Westphal’s sign in the SCO group (3.0 (1.0–5.0) min) was longer than that in the SPI group (0.5 (0.5–0.5) min) (*p* = 0.009, CI95%: 0.62–3.38), DPE (0.8 (0.5–2.0) min) (*p* = 0.011, CI95%: 0.38–3.12) and LS groups (0.5 (0.5–3.0) min) (*p* = 0.009, CI95%: 0.52–3.26) (Figure 3).

The Kaplan–Meier analysis showed that, without controlling any other factor, the time to Westphal’s sign was significantly different for all the study groups (*p* < 0.001) (Figure 4).

All the neuraxial blocks were considered successful by the attending anaesthetists.

## 4. Discussion

All the neuraxial techniques performed in this study were successful both in the suppression of the patellar tendon reflex and in the opinion of the attending anaesthetists of the cases. There was no difference in the number of attempts to perform these techniques, nor in the time to perform them (although both epidural injection groups tended to be slightly shorter than the others). Finally, the time to the suppression of the patellar tendon reflex (Westphal’s sign) was longest in the SCO group, being the shortest in the SPI group.

The Westphal’s sign refers to the absence of or decrease in the patellar tendon reflex, also known as knee-jerk reflex, in the physical exam [20]. The patellar tendon reflex is the kicking-like response of the knee after the stimulation of the patellar tendon. This stimulation, traditionally performed by a reflex hammer, stretches the muscle spindles located in the quadriceps muscles, which excites the afferent sensory neuron. This afferent sensory neuron synapses on an efferent α motor neuron (monosynaptic reflex) that contracts the quadriceps muscle [21]. Both the afferent sensory and the α motor neurons are part of the femoral nerve, originating at the L4-L6 level of the spine in the dog [22].

An absent or diminished patellar reflex may be due to a pathology or block affecting the afferent sensory neurons or the efferent motor neurons. In general, if the reflex is absent in conjunction with sensory loss, the lesion is likely to be in the afferent sensory neurons. If the reflex is absent with concurrent paralysis, muscle atrophy, or muscle fasciculations, the lesion is likely to be in the efferent motor neurons [23]. In the case of regional anaesthesia such as spinal and epidural injections as in this study, it is thought that the alteration occurs at the level of the spinal cord in the synapses between both neurons. This way, the stimulus from the stretched muscle reaches the spine, but it cannot be transmitted to the efferent motor neurone to induce the kicking-like movement. For this reason, the reflexes are altered after the initiation of an epidural or spinal block [24]. In general, the time to Westphal’s sign was relatively quick in all dogs. The longest time obtained in the study was only five minutes in a dog in the SCO group, being the time of the entire group 3.0 (1.0–5.0) min. This time was very similar to the onset time previously reported in the literature in dogs (mean ± SD of 3.75 ± 1.76, 4.1 ± 2.5, and 3.2 ± 2.0 min) [25,26,27]. Unfortunately, the direct comparison between these studies and the research presented here is not straightforward. In this study, the assessment of the femoral nerve was performed using the time to Westphal’s sign, while Lebeaux, Feldman and Covino, and Feldman et al. all used the time to ‘loss of weight support’ [25,26,27]. However, taking into consideration the differences in the method of assessment, both studies studied the effect of bupivacaine on the femoral nerve (L4 to L6 spinal nerves) and its onset of action.

The groups DPE and LS had a significant overlapping between them and with the SPI group in the time to the Westphal’s sign. In the Kaplan–Meier ‘survival’ curve, it can be observed that at least 50% of the animals in the DPE and LS groups had no patellar tendon reflex present by the time of the first assessment at 0.5 min. This is easily explainable in the DPE group, as the technique aims for migration of the local anaesthetic through the previously created orifice in the dura mater, so the patient can benefit from the quick onset of action and longer duration [15]. However, the results cannot be so easily explained in the case of the LS group. In 2015, Liotta et al. reported that 81% of cadavers (13 out of 16) receiving a lumbosacral epidural injection of contrast medium through a spinal needle presented some degree of subarachnoid contamination when analysed with a CT scan. When the same protocol was repeated in live dogs, all five dogs that were attempted presented subarachnoid contamination [28]. Additionally, in this finding, there is an increasing number of articles demonstrating the presence of dural sac (and consequently subarachnoid space) at the lumbosacral intervertebral space in dogs. For instance, Lang and Marin-Garcia et al. reported the end of the dural sac at the level of the first-second and second-third sacral vertebrae in medium to large dog breeds and German Shepherd dogs, respectively [7,29]. Zapata et al. reported the termination of the dural sac caudal to the lumbosacral intervertebral space in more than 50% of all dogs studied with myelography and in more than 75% of small dogs (<10 kg) [30]. When scrutinising magnetic resonance images of a multitude of dogs a various breeds and sizes, Khan et al. found that the most common dural sac termination in dogs was in the cranial half of the sacrum, irrespective of bodyweight or spinal cord length [31]. Regardless of the exact point at which the dural sac ends, the presence of this sac was evident at the lumbosacral intervertebral space in a great number of dogs of various sizes and breeds, to the point that some of these authors warned the reader about the risk of performing a subarachnoid puncture when attempting an epidural injection [30].

Despite not finding significant differences between the time taken to perform the different neuraxial techniques, the groups SPI and DPE tended to be longer than the LS and SCO groups. The time to perform the epidural injection in the SCO group (46 (26–210) s) was very similar to the one previously reported (40 (19–137) s), also using the nerve stimulator to identify the epidural space at the sacrococcygeal interverbal space [11]. The time for the LS group (34 (10–120) s) was also relatively similar to the time reported to perform epidural injection in the lumbosacral space with dogs in sternal recumbency (20 (14–79) s) [32]. Unfortunately, there is no previously reported information regarding the time taken to perform SPI injections in dogs, so the results from this study cannot be put in context with rest of the literature. This is the first study exploring the DPE technique in dogs, and the time recorded to complete both injections was quite variable; however, the similarity with the SPI time suggests that the limiting factor and more time-consuming phase for this technique is the dural puncture. It is understandable that these techniques may be more time-consuming, as they can be considered not commonly performed or ‘mainstream’ in veterinary anaesthesia. With that in mind, and taking into consideration that a spinal injection refers to the placement of the tip of a needle just millimetres away from the central nervous system, the longest time recorded in this study (447 s (just under 7.5 min) in a dog in the SPI group) does not look unreasonable for a successful uncomplicated locoregional technique.

This study presents several limitations in its design. Firstly, the time to the Westphal’s sign was considered the positive endpoint and a surrogate of the local anaesthetic ‘block’. Although radiographic studies [10] or the lack of response to the surgical insult [32] could have been used to assess the success of the different techniques, it was paramount for the design of this study that the assessment was repeated at close and regular intervals to be able to determine the onset of action of these techniques. The patellar ligament reflex is an innocuous and repeatable test that can be and has previously been used in anaesthetised animals to assess the success of epidural injections [11,12], and for that reason, it was chosen as the most suitable method of evaluation. Secondly, a combined spinal–epidural injection group was not included in this study. This technique has been used extensively in human anaesthesia since the 1980s and involves the delivery of drugs in the subarachnoid space, followed by another administration epidurally. Theoretically, this technique aims for a rapid onset of action from the spinal injection with a prologue duration of action from the epidural one [33]. The study presented here assessed the onset of action of the different techniques, but not their duration (fundamentally due to the different types of surgeries used). During the preliminary discussions before the study design was finalised, it was considered that the inclusion of a combined spinal–epidural group would match closely the SPI group onset of action without showing any further difference (as the duration of the block was not going to be assessed). Due to this, and not wanting to complicate the study by using two novel techniques (with great similarities between them) (DPE and combined spinal–epidural injection) and one technique not commonly used in practice (SPI), this group was not added to the study design. Thirdly, the DPE technique described in human anaesthesia uses a variety of needle insertions from needle-through-needle to separated needles inserted in the same intervertebral space or in different intervertebral spaces. The variation used here was selected as a combination of techniques that were familiar to the operator, but the results obtained here might have been different if another needle insertion technique had been used. Finally, the time to perform the neuraxial techniques exclusively included procedure time, not counting animal preparation (eg. hair clipping, skin preparation, changing of recumbency, etc.), draping, or instrument preparation. From the developing phases of this study, it was decided that the time to complete a procedure (independently if this is a long surgical procedure or a few minutes long local anaesthetic technique) should only depend on the complexity of the procedure itself and the ability of the person performing it. It is for that reason, and to allow fair comparison between groups, that the stopwatch was stopped during the two phases of the DPE technique. This allowed the comparisons that we already described above, but any further interpretation of these data must be cautious, as these procedure times were shorter than the clinically relevant time elapsing from the beginning of the hair clipping until the end of the bupivacaine injections. To allow some contextualisation, the preparation of any animal for neuraxial anaesthesia never took more than five minutes, and the change in recumbency and alcohol spray of the lumbosacral area during DPE was typically completed in less than one minute.

## 5. Conclusions

All four techniques were successful in abolishing the patellar ligament reflex (Westphal’s sign) rapidly. The epidural injections at the sacrococcygeal intervertebral space were the slowest in producing this effect, while the lumbosacral epidural, spinal, and dural puncture epidural injections were faster and almost indistinguishable in the patellar reflex block onset time.

## Figures and Tables

**Figure 1 animals-11-02996-f001:**
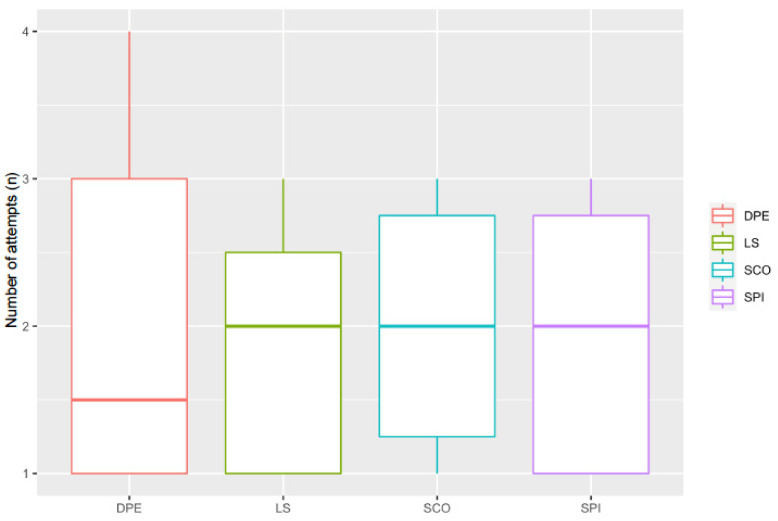
Number of attempts (*n*) required to perform each technique. DPE, dural puncture epidural injection; LS, epidural injection in the lumbosacral space; SCO, epidural injection in the sacrococcygeal space; SPI, injection in the subarachnoid space. No difference was found between the groups.

**Figure 2 animals-11-02996-f002:**
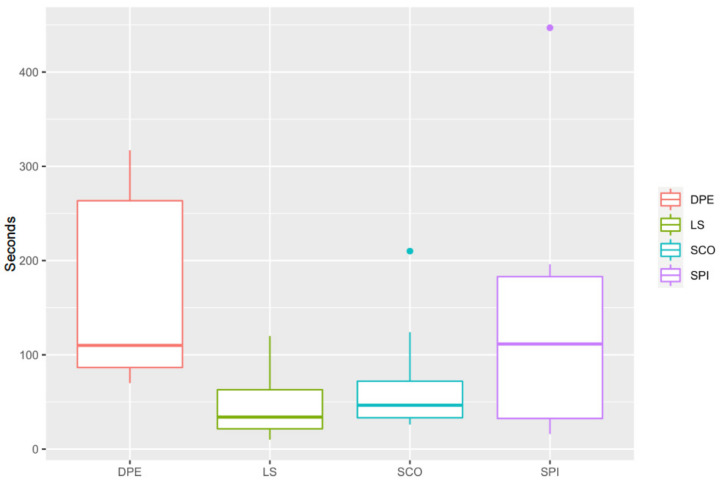
Time to perform the technique (not counting with the time for drugs administration) in seconds. DPE, dural puncture epidural injection; LS, epidural injection in the lumbosacral space; SCO, epidural injection in the sacrococcygeal space; SPI, injection in the subarachnoid space. No difference was found between the groups.

**Figure 3 animals-11-02996-f003:**
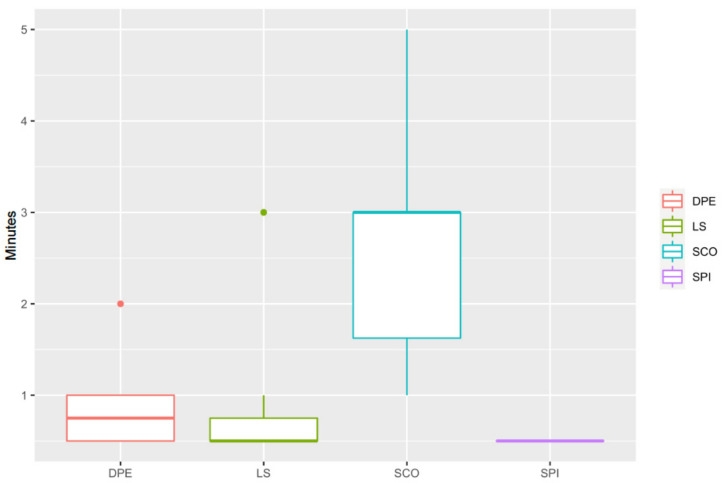
Time to the abolishing of the patellar ligament reflex or Westphal’s sign in minutes. DPE, dural puncture epidural injection; LS, epidural injection in the lumbosacral space; SCO, epidural injection in the sacrococcygeal space; SPI, injection in the subarachnoid space. SCO was significantly slower in blocking this reflex than the rest of the groups.

**Figure 4 animals-11-02996-f004:**
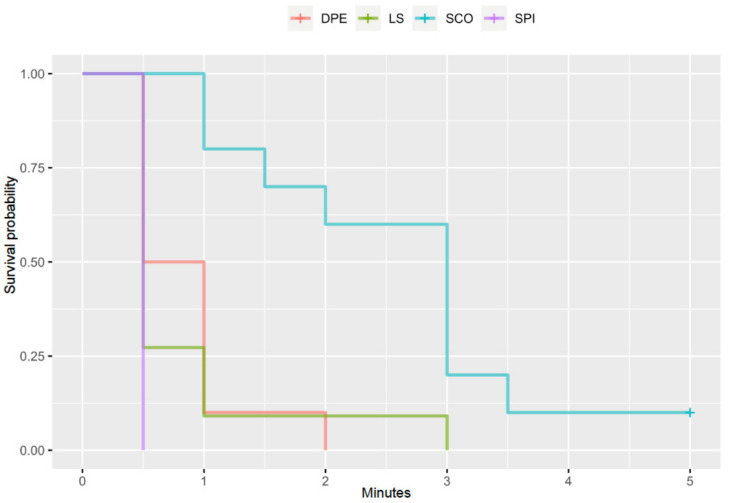
Kaplan–Meier curve of time to event probability of disappearance of the patellar ligament reflex or Westphal’s sign in 41 dogs receiving neuraxial anaesthesia. DPE, dural puncture epidural injection; LS, epidural injection in the lumbosacral space; SCO, epidural injection in the sacrococcygeal space; SPI, injection in the subarachnoid space.

**Table 1 animals-11-02996-t001:** Numbers (*n*) of surgical procedures performed in each of the study groups. SCO, epidural injection in the sacrococcygeal space; SPI, injection in the subarachnoid space; LS, epidural injection in the lumbosacral space; DPE, dural puncture epidural injection.

		Groups (*n*)			
		SCO	SPI	LS	DPE
Types of surgeries	Tibial plateau leveling osteotomy (TPLO)	4	4	3	3
	Fracture repair	2	1	1	
	Total hip replacement	1		1	1
	Patella luxation repair		3		2
	Mass removal ± skin flap	2		3	
	Leg amputation			1	1
	Arthroscopy	1	2		2
	Arthrodesis			1	1
	Achilles’ tendon rupture repair			1	
	Total	10	10	11	10

**Table 2 animals-11-02996-t002:** Sex distribution and mean ± SD of weight, age, body condition score (BCS), occipital-coccygeal length (OCL), total volume of local anaesthetic injected in the neuraxial technique, and the number of dogs per group. SCO, epidural injection in the sacrococcygeal space; SPI, injection in the subarachnoid space; LS, epidural injection in the lumbosacral space; DPE, dural puncture epidural injection.

	Groups				*p*-Value
	SCO	SPI	LS	DPE	
Sex: Male/Female (*n*)	1/9	8/2	4/7	3/7	0.01
Weight (kg)	21.0 ± 20.8	19.8 ± 11.5	14.0 ± 7.8	16.1 ± 8.1	0.38
Age (months)	94 ± 38	66 ± 50	82 ± 43	110 ± 47	0.49
BCS (1 to 9)	5 ± 2	4 ± 1	5 ± 2	5 ± 2	0.39
OCL (cm)	63.5 ± 22.7	63.5 ± 15.4	55.9 ± 16.1	62.7 ± 10.1	0.37
Total volume of local anaesthetic injected (mL)	3.5 ± 1.4	1.3 ± 0.6	2.9 ± 0.9	3.3 ± 0.6	0.57
Total number of dogs (*n*)	10	10	11	10	

**Table 3 animals-11-02996-t003:** Distribution of the different epidural drug combinations chosen by the attending anaesthetists, where (*n*) is the number of dogs per group. No statistical analysis was possible/performed. SCO, epidural injection in the sacrococcygeal space; SPI, injection in the subarachnoid space; LS, epidural injection in the lumbosacral space; DPE, dural puncture epidural injection.

Drug/s	Groups (*n*)			
	SCO	SPI	LS	DPE
**Preanaesthetic medication**				
Medetomidine + Methadone	4	3	5	3
Dexmedetomidine + Methadone	4	1	1	2
Methadone	2	2	1	1
Acepromazine + Medetomidine + Methadone		3	2	4
Medetomidine + Methadone + Ketamine		1		
Acepromazine + Methadone			2	
Total	10	10	11	10
**Induction agent/s**				
Propofol	3	3	6	4
Alfaxalone	5	4	4	6
Propofol + Ketamine	1	2	1	
Ketamine + Midazolam		1		
Alfaxalone + Midazolam	1			
Total	10	10	11	10

## Data Availability

The entire dataset is available upon request to the correspondence author.

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
