# Peer review of "Onset of Action of Bupivacaine Administered via Dural Puncture Epidural versus Spinal, Lumbosacral, and Sacrococcygeal Epidural Injections in Dogs: Randomised Clinical Trial"

_animals, 2021, doi:10.3390/ani11102996_

Round 1
Reviewer 1 Report
Dear Authors
thank you for submitting this nice paper, In the attachment you can find my comments
Best Regards

Reviewer 2 Report
This interesting study evaluates 4 different neuraxial techniques in dogs, using bupivacaine and buprenorphine and assessing a novel technique in veterinary as dural puncture epidural (DPI). It is a well-designed and organized study and clear in its description. The objective of comparing the action times of the techniques is achieved, although certain issues must be improved. The authors should review the way in which they count the times in some techniques, as this would influence the results obtained. Although the authors did not have differences in the times, they base the conclusions on trends, and this is not correct, but it could be affected if the criteria of the times are modified.
Although it is not a primary objective of the study, basing the consideration of the efficacy of the technique on the appreciation of the anesthetist is somewhat subjective and it would be advisable to provide more information. Other comments are indicated below.
- Please consider “technique” to “procedure or method”.
77-78. What is the reason to hypothesize that epidural would be faster than spinal anaesthesia?
- At what point prior to the neuraxial technique was the animal's reflex checked?
107-109. Was it not possible to detect hemodynamic events after the administration of neuraxial techniques, or did they not occur in any patient?
- What is considered by authors as "stabilization"?
- In the manner as authors describe the clipped area in each group, to the researcher could be possible to difference between epidural and spinal groups. At least, between SPI and the rest. I understand that this is the reason to cover it with a blanket and this explanation should be provide to readers.
179-180. It should be clear that the time of action of bupivacaine goes from neuraxial injection until the patellar reflex disappears.
181-182. If the response was considered positive, why did authors measure again 30 sec later? The logical thing is to consider positive when the reflection is totally absent, which is when it really begins to take effect. In the results section do not indicate the number of patients who required a second evaluation. This could influence the results of the times.
- Since the technique starts when needle rests on skin and finished when drugs are delivered, the time proposed by authors would be shorter than real time. I know that other studies consider the time like authors, but it is not real. Tt is more logical to consider the time based on the full duration of the technique.
189-191. This information should appear previously after “target (either epidural or subarachnoid space) was reached”. What is the reason to stop the chronometer? The DPE technique is the combination of both maneuvers, and the time should be continuous, if not, the actual technique time is being falsely reduced.
191-193. I can understand that this decision on DPI was to avoid significant differences to other techniques in the number of attempts (a minimum of two attempts would be needed) but it is not clear. It can be confusing for readers because authors only considered the number of attempts in the maneuver with the highest number and not the total attempts (both of them). So please clarify.
198-201. This information is highly subjective.
205-208. Authors should be provided SD and/or effect size to check the size sample determination.
210-213. The authors do not report whether they perform a normality test on the data but considering that they are a small number of animals per group, why did they not consider performing a non-parametric test analysis? (ex. Kruskall-Wallis test). Especially to the number of attempts.
- Please consider change median (range) to median (P25th-P75th).
- If dogs were randomized into four study groups electronically, and there were no losses in the research, why were 41 animals used and not 40 (10 per group)?
223-224. I am not sure if the authors did not make a statistical comparison between the characteristics of the groups. If so, the results should be provided (226-227). The authors state that the only difference was detected in the distribution of males but looking at the data on the total volume of bupivacaine, significant differences between groups should be detected (something logical using different doses). Please clarify.
229-230. The value of the median in DPE, is it 1.5 or 2?
230-233. I assume that the absence of significant differences was due to that criterium of authors to define the time of administration. This result supports my view that time should be considered in its entirety. It is no logical that a technique (DPE) with two maneuvers has a similar time to others with only one. Also, it should be considered the influence on this (absence of differences) of the potential effect of outliers in groups of low number of animals, especially in SPI (447 minutes).
235-236. According to Kaplan-Meier curve I believe that mean of Westphal’s sign in the LS group should be higher 0.5. Please check the values.
234-236. It would be highly advisable to provide the median (CI95%) of the differences.
- It could be useful to readers to know if rescue analgesia was considered in the clinical cases. The anesthesiologist's assessment alone seems subjective to consider the success of the techniques. It is an exceptional value that different neuraxial techniques (even some as rare as DPE) have 100% success in clinical conditions, even with very proximal interventions of the hind limbs. Considering also that they are very varied and even long surgeries, where the loss of effect of the techniques cannot be ruled out.
Figure 4. Although authors indicate that the dog with the longest time to lose the patellar reflex in group SCO was at 5 minutes, in the graph this value appears truncated.
- There is a paragraph break. Please correct it.
Reviewer 3 Report
Dear authors,
Thank you for submitting an well-designed clinical study. I find your work interesting and well-presented. My only comment is on the data analysis. Were the data checked for normality? From the box-plots, I doubt that they were normally distributed (I can see some extreme values or outliers). So, I do not believe that ANOVA is the most appropriate test to analyse the data.
And a minor correction: In line 168, please replace "injection" with "puncture". As you mention, there was no drug administration, so "injection" is misleading.
Round 2
Reviewer 2 Report
I appreciate the responses that the authors have made to my questions and the changes they have made to the article. This is a very interesting study. Although most of the issues have been resolved, there are still some limitations that should be corrected.
I understand the criteria of the authors to establish the number of attempts (the highest). However and although I understand the point of view of authors in relation to times and previous studies, but it does not seem reasonable that there are no time differences between single-stage and two-stage procedures (DPE). Something that although non statistically relevant, from a clinical and real point of view it is.
As it is a procedure that involves two phases and two positions, this must inevitably affect time. It would be from the moment in which the needle is inserted in the desired position, until the procedure is completed with the administration of bupivacaine. If the authors want to maintain their position, they should include in the limitation´s section that the absence of differences could be influenced by this criterion. I regret to be demanding in this regard, but in general, the most relevant results of the comparison between the techniques in this study are number of attempts, efficiency ((Westphal’s sign) and time, and for this reason I believe that it is necessary to be more precise.
